# One Health Disparities and *Blastocystis* infection among smallholder farmers in northeastern Madagascar

Alma Solis[1,2]*, Angela Anaeme[1], Georgia Titcomb[3], Mark Janko[2],
Jean Yves Rabezara[4], Tyler M. Barrett[1], Kayla Kauffman[5], Michelle Pender[2],
Voahangy Soarimalala[6,7], Lev Kolinski[1], Randall Kramer[2,8], Hillary Young[5],
Charles L. Nunn[1,2]

**1** Department of Evolutionary Anthropology, Duke University, Durham, North Carolina, United States of America, **2** Duke Global Health Institute, Duke University, Durham, North Carolina, United States of America, **3** Department of Fish, Wildlife, and Conservation Biology, Colorado State University, Fort Collins, Colorado, United States of America, **4** Science de la Nature et Valorisation des Resources Naturelles, Centre Universitaire Régional de la SAVA, Antalaha, Madagascar, **5** Department of Ecology, Evolution, and Marine Biology, University of California Santa Barbara, Santa Barbara, California, United States of America, **6** Association Vahatra, Antananarivo, Madagascar, **7** Institut des Sciences et Techniques de l'Environnement, University of Fianarantsoa, Fianarantsoa, Madagascar, **8** Nicholas School of the Environment, Duke University, Durham, North Carolina, United States of America

* alma.solis@duke.edu

## Abstract

*Blastocystis* is a globally transmitted gastrointestinal protozoa that commonly infects people living in low- and middle-income countries. Transmission is thought to occur via consumption of contaminated water or food and through contact with infected animals, although the specific factors that influence infection in low-resource, rural settings remain unclear. We applied a One Health Disparities framework that considers the interconnectedness of human, nonhuman animal, and environmental health to investigate disparities in *Blastocystis* spp. infection in rural northeastern Madagascar. We focused on a suite of predictors including wealth, animal contact, hand hygiene, and demographic factors. Overall, 76.5% of 783 participants were infected with at least one of three subtypes of *Blastocystis*, and 19% of people were co-infected with two or more subtypes (ST1, 2, and 3). We found that men had lower risk of infection than women, while individuals who reported washing their hands without soap had higher odds of infection across all subtypes. Within a single subtype, soap-use remained significant for both ST1 and ST2, while for ST3, the effect of gender remained significant. Wealth and animal interactions had no significant associations with infection. Our study sheds light on gender disparities and the importance of hand hygiene in explaining variation in *Blastocystis* infection in rural Madagascar, while failing to support hypotheses based on socioeconomic status and exposure to domesticated animal reservoirs of disease. The findings also underscore the importance of gastrointestinal infections in vulnerable rural populations in Madagascar and highlight

**Data availability statement:** The data that support the findings of this study are available on request from the corresponding author. The data are not publicly available due to privacy or ethical restrictions. Participants did not give their consent to have individual level data made public. Data requests can be made to Duke University Campus Institutional Review Board (campusirb@duke.edu) with Holly Williams (holly.williams.irb@duke.edu) being the point of contact.

**Funding:** This work was supported by the joint NIH-NSF-NIFA Ecology and Evolution of Infectious Disease Program (Grant No. R01-TW011493 to CLN) and the National Academies of Sciences, Engineering, and Math Ford Foundation Predoctoral Fellowship (to AS), and the Duke Global Health Institute Doctoral Scholar Program (to AS). The funders had no role in study design, data collection and analysis, decision to publish, or preparation of the manuscript.

**Competing interests:** The authors have declared that no competing interests exist.

ways to address health equity and environmental justice in rural, low-resource settings.

## Introduction

*Blastocystis* is a globally distributed, enteric protozoan that infects humans and a wide range of non-human animals (hereafter, "animals") that include mammals, amphibians, reptiles, and arthropods [1,2], making it one of the most common gastrointestinal protozoa [2]. Approximately 1 billion people worldwide are infected with *Blastocystis* [3], although prevalence is especially high in low-income countries [1,4]. Research on *Blastocystis* infection is concentrated in Europe, Asia, and Latin America, and data on prevalence and diversity in Africa remains limited, leaving critical questions about prevalence of infection and transmission dynamics in rural settings unanswered [5].

Identifying *Blastocystis* infection is challenging due to its wide range of symptoms and high rate of asymptomatic infections. *Blastocystis* infections are often asymptomatic, though symptoms can include gastrointestinal distress such as diarrhea, abdominal pain, vomiting, excessive gas, and bloating [6–8]. The variation in symptoms may be partially attributed to the broad genetic diversity of the parasite [8,9], which arises from *Blastocystis'* wide geographic and host distributions. To date, 28 unique subtypes (ST) have been identified [10,11], of which 15 are associated with both human and animal hosts [12]. STs 1–4 are commonly reported in people, with ST3 being the most common, followed by ST1 [7, 11].

Determining the transmission route of *Blastocystis* is complicated by this genetic diversity and the range of potential factors associated with physical, biotic, and social environments. Commonly proposed risk factors for *Blastocystis* infection include consumption of contaminated water and interactions with infected animals [4,13]. Due to the risk of transmission via consumption of contaminated water, limited access to adequate hygiene infrastructure may also influence transmission [14]. Infected individuals shed *Blastocystis* in feces, providing opportunities for transmission via the fecal-oral route [3,8,15] and diagnosis is made through stool samples [8]. Due to variation in symptoms and lack of testing, the prevalence and burden of *Blastocystis* is likely underestimated, especially in low-resource settings where health authorities lack the resources to diagnose infection [16].

Understanding *Blastocystis* transmission requires considering interconnected factors among humans, animals, and the environment, a concept embodied in the One Health approach [17].The One Health Disparities (OHD) framework adds another dimension by integrating the human social environment [18]. While One Health highlights the interconnectedness of human, animal, and environmental health [19], it often overlooks the sociocultural environment's impact on human-animal-environment interactions [20]. Addressing these gaps is crucial for mitigating disease because factors like access to clean water, sanitation, and healthcare play a significant role in disease risk [20]. Similarly, sociocultural factors influence human-animal interactions,

such as pet ownership, raising livestock, food handling, and consumption [20]. This study applies the OHD framework to investigate variation of a gastrointestinal pathogen, *Blastocystis*, infection among smallholder farmers in northeast Madagascar, focusing on hypotheses that assess characteristics of wealth, hygiene practices, and human-animal interactions.

Farmers in northeast Madagascar have a high level of animal interaction, variable access to clean water, and limited availability of high-quality hygiene products like soap, all of which may influence the risk of *Blastocystis* infection. Rural populations in low- and middle-income countries (LMICs) often have limited access to healthcare compared to more readily available health resources in urban settings [21,22]. In addition, access to improved housing materials, including floors that are constructed from purchased materials rather than naturally sourced materials, are associated with lower odds of infection with enteric parasites [23], potentially because earthen walls or floors may be more hospitable environments for parasite survival. Moreover, low socioeconomic status (SES) is linked to food insecurity and malnutrition, which can increase susceptibility to infectious diseases [24]. Overall, low SES is a key determinant of health associated with increased morbidity and mortality [25].

To improve understanding of *Blastocystis* transmission and OHD in low-income settings, we investigated three hypotheses. First, we investigated the hypothesis that lower SES is associated with increased infection risk due to limited access to healthcare and safe housing infrastructure. Second, we investigated whether people acquire *Blastocystis* from animals by testing the prediction that people who report higher rates of animal interactions exhibit higher risk of *Blastocystis* infection. This hypothesis is based on the wide host range of *Blastocystis* and the close contact between humans and other animals in our study population. Finally, we investigated the hypothesis that people's understanding and education of disease transmission impacts infection risk, predicting that poorer hygiene practices and limited understanding of zoonotic diseases is associated with greater risk of *Blastocystis* infection. In addition to the variables listed above, we investigated age and gender as demographic factors that explain variation in *Blastocystis* infection.

## Methods

### Ethical statement

The Institutional Review Board of Duke University (IRB: 2019–0560) and the Malagasy Ethics Panel (Permit Number 114 MSANP/AGMED/CERBM) approved the protocol used in this study. Surveys were conducted in the local Malagasy language by trained research assistants, and survey responses were recorded on Samsung tablets using Qualtrics software [26]. All survey participants were 18 years of age or older and provided oral and written informed consent and documented in Qualtrics survey and witnessed by our trained research team. Participants were compensated with pre-paid mobile phone credits. Recruitment occurred between October 3rd, 2019, and August 8th, 2022, using snowball sampling, in which surveys included name generators to create lists of individuals to recruit for subsequent surveys, as described by Kauffman and colleagues [27].

### Study area and population

Data collection occurred between 2019 and 2022 across three villages in northeastern Madagascar on the periphery of Marojejy National Park. Data collection was temporally distributed to align with different rainy and dry seasons as part of a larger study. For community privacy, hereafter the village sites will be described as Village A, Village S, and Village M. The three villages are small-scale farmer communities with varying population sizes, ranging from 900 to 2,700 people.

### Survey Questions

We developed a survey instrument to capture wealth, animal contact patterns, and hygiene practices (S1 Fig). To test our first hypothesis related to wealth, we used two proxy numerical variables to assess an individual's wealth. To collect the two proxy wealth variables, we first asked three questions about the material used for a household's outer wall, roof,

and floor. The three questions had the same options: thatch, compacted earth, burnt brick, metal sheets (iron, aluminum), mud, mud brick, concrete/cement, bamboo, wood planks, travelers palm, raffia palm. We excluded participants who selected "other" due to the insufficient number of individuals in that category. The material used for each of the three categories was numerically ranked from 1 to 4 based on whether the materials could be sourced locally and their costs [26]. We then summed across the three questions to create the housing material index score, potentially ranging from 3 to 12, with a higher score indicating houses that were constructed using more costly materials (S2 Fig). The second wealth index variable captured the sum of commercial goods and land owned. For this, we asked participants whether their household owned any of the following durable goods: mobile phone, radio/CD player, television, computer, refrigerator, generator, bicycle, motorcycle/scooter, car/truck, boat, animal drawn cart [26,28]. We also asked if they owned land, scored as a binary variable. We then summed across all responses to create the sum of goods and land owned index score, potentially ranging from 0 to 12.

To test our second hypothesis involving human-animal interactions, we used two numerical variables to measure the sum of animal contact and sum of all animal interactions [29]. First, we asked questions about eight types of animal interactions: owning a pet, seeing an animal inside their dwelling, handling live animals, raising an animal, eating raw or undercooked animal tissues, eating an animal that was sick, sharing a water source with animal(s) for washing, and seeing animal feces in or near their food prior to eating. These questions further requested participants to specify the animals they interact with, drawing from 11 categories: rodents/shrews, tenrecs, wild birds, carnivores, poultry, goats/sheep, domestic pigs, bush pig, cattle, dogs, or cats (S1 Fig).

From the eight questions, we created two numerical variables to measure human-animal contact. First, we calculated the total number of animal species with which a person interacted, to account for interactions with more than one animal, which hereafter is called *animal richness* and had a potential range from 0 to 11. Second, we calculated the total number of types of animal interactions reported by a participant, which we hereafter call *animal interactions*, with a potential range from 0 to 8.

To assess hygiene practice and zoonotic disease literacy we incorporated two variables. First, to create the *Zoonoses Awareness* variable, we asked if participants were aware of human-animal disease transmission, scored as a binary (yes/no) variable. In addition, to create the *Handwashing Practice* variable, we asked how they typically wash their hands, with the following responses as options: soap, ash, sand, water only, or do not wash hands. No respondents reported using ash, sand, or none.

Socio-demographic characteristics included in all our models were gender measured as a binary categorical variable (M/F), age (years), and education as a binary categorical variable scored as either "primary education or less" versus "secondary education or higher," and village.

### *Blastocystis* DNA extraction and sequencing

Fecal samples were collected from the same consenting survey participants and transferred into a 2ml Eppendorf tube with Zymo DNA/RNA Shield or nucleic acid preservation (NAP) buffer [30] then stored at room temperature for up to 7 days before being transferred to the Duke Lemur Center office in Sambava and stored at -20°C.

To detect *Blastocystis* among survey participants, DNA was extracted from fecal samples using Zymo Miniprep Fecal Kits (Zymo Research, Irvine, California) and DNA was amplified, using polymerase chain reaction (PCR) Guanine-quadruplex (G4) primer set for protozoan organisms, which targets the hypervariable regions of the 18S ribosomal RNA [31,32]. PCR reactions were performed with dual-indexed barcodes of 8 nucleotide primers carried out in 15μL volumes that consisted of the following: 3μL of forward primer (2mM stock concentration), 3μL of reverse primer (2mM stock concentration), 2μL of template DNA, and 7μL of Mastermix (0.7μL of Amplitaq Gold polymerase, 1μL MgCl2, 150μL Amplitaq Gold buffer, 12μL BSA, 6μL DMSO, and 344μL water). PCR cycling conditions for the G4 primer set were: 10min hot-start activation, 35x cycles of 15s at 95°C, 30s at 57°C, 40s at 72°C, with a final 5-minute extension at 72°C.

After amplification, final DNA concentrations were measured using Promega One Quantifluor kits on a Tecan plate reader. PCR product concentrations were adjusted to 7ng/mL before pooling, and the product was cleaned using Min-Elute columns and magnetic bead cleanup with a bead:DNA ratio of 8:10. The final libraries were multiplexed together before sequencing at the UC Davis Genome Center, and Illumina adapters were ligated to the final library. For sufficient read depth, three repeated sequencing runs on an Illumina MiSeq (v3 2x300bp, 25M reads) were performed, with a PhiX spike-in of 15% per run.

Individual samples were then demultiplexed using *cutadapt* with zero error tolerance [33]. Further filtering steps were then performed using the *dada2* R package workflow [34] to remove errors, dereplicate, trim and filter amplicons (minimum length = 100, 15% PhiX removed), merge pairs, and finally determine amplicon sequence variants (ASVs) with the pseudo-pooling method [32]. The relative read abundance of each ASV in each sample was calculated. To avoid potential sequencing errors or tag jumps, ASVs that had less than 2% of a sample's relative read abundance were removed.

To assign taxonomy to all ASVs, we compared results from the assignTaxonomy function in dada2 to the IdTaxa function in the DECIPHER package in R to identify consensus *Blastocystis* IDs [35]. We then used the NCBI GenBank database identify *Blastosystis* STs using the blastn function. We calculated the mean percent identity for each taxonomic identification derived from the top 100 hits and the ST with highest percent identity was assigned to each ASV.

## Statistical Analysis

All statistical analyses were performed using R version 4.3.2 [36]. We fit five generalized linear models (GLMs) to test our hypotheses, with the following response variables: (a) *Blastocystis* infection with any subtype as a binary response (0,1), (b) ST richness (sum of infections with ST1, ST2, and ST3) as a numerical response, and (c) infection (0,1) with, and only with, each of the subtypes ST1, ST2, and ST3 (i.e., we excluded participants who were coinfected with 2 or more ST infections). We focused on individuals with only one ST to avoid any interactions that might occur among sub-types in co-infection. All five models included the same predictive variables (e.g., Response Variable ~ Gender + Age + Education + Housing Material + Sum of Good Owned + Handwashing + Zoonotic Literacy + Animal Interactions + Animal Richness + Village). For the binary-response models, we used GLMs fit to binomial distributions to predict the odds of *Blastocystis* infection. For analyses of ST richness, we fit the GLMs to a Poisson distribution to predict the number of infections and co-infections with ST1, ST2, and ST3. Using the GLM coefficients, we calculated odds ratios (OR) for the logistic regressions and rate ratios (RRs) for the Poisson regressions, as well as 95% confidence intervals (CI) and standard p-values ≤ 0.05 to determine statistical significance.

To evaluate the statistical models, we first conducted a multicollinearity analysis among the predictor variables using the *car* package [37], using the Variance Inflation Factor (VIF) score. The model's overall fit was evaluated using Pseudo R-squared values using the *pscl* package [38] and for the Poisson regression model, we further assessed the model for overdispersion by calculating the dispersion ratio using the *MASS* package [39].

## Results

### Participant characteristics and *Blastocystis* screening

We sampled adult participants from villages A, S, and M. Table 1 provides a summary of socio-demographic details and variation in predictor variables used to test hypotheses). We had a final sample size of 776 participants, reflecting the number of individuals with no missing data across the variables. *Blastocystis* screening was conducted on all 776 fecal samples from participants across three villages: Village M (n = 171), Village S (n = 220), and Village A (n = 385) (Table 1). Overall, 583 (76.5%) participants were infected with *Blastocystis*, and we identified three main subtypes: ST1, ST2, and ST3 (Fig 1). Of the infected people, 19% (n = 147) were co-infected with two or more subtypes.

**Table 1. Demographic characteristic and survey summary responses across the three villages.**

| | Total N = 776[1] | A N = 385[1] | S N = 220[1] | M N = 171[1] |
|---|---|---|---|---|
| **Gender** | | | | |
| Female | 370 (48%) | 191 (50%) | 96 (44%) | 83 (49%) |
| Male | 406 (52%) | 194 (50%) | 124 (56%) | 88 (51%) |
| **Age** | | | | |
| | 34 [25,49] | 34 [25,48] | 30 [23,46] | 40 [31,54] |
| **Education** | | | | |
| ≤Primary Education | 386 (51%) | 179 (47%) | 96 (44%) | 111 (65%) |
| >Primary Education | 390 (50%) | 206 (54%) | 124 (56%) | 60 (35%) |
| **Handwashing** | | | | |
| Water only | 348 (45%) | 158 (41%) | 73 (33%) | 117 (68%) |
| Soap and Water | 428 (55%) | 227 (59%) | 147 (67%) | 54 (32%) |
| **Zoonoses Awareness** | | | | |
| Yes | 591 (76%) | 308 (80%) | 162 (74%) | 121 (71%) |
| No | 185 (23%) | 77 (20%) | 54 (25%) | 50 (29%) |
| **Housing Material Score** | | | | |
| | 6.00[4.00,8.00] | 6.00[4.00,8.00] | 7.00[5.75,8.00] | 5.00[4.00,8.00] |
| **Sum of Goods Owned** | | | | |
| | 3.00[2.00,4.00] | 3.00[2.00,4.00] | 3.00[2.00,4.00] | 3.00[2.00,4.00] |
| **Animal Interactions** | | | | |
| | 5.00[4.00,6.00] | 5.00[4.00,6.00] | 5.00[4.00,6.00] | 5.00[4.00,6.00] |
| **Animal Richness** | | | | |
| | 5.00[4.00,6.00] | 5.00[4.00,5.00] | 5.00[4.00,6.00] | 5.00[4.00,6.00] |

[1]n (%); Median [IQR].

## Predictors of *Blastocystis* infection

The binomial logistic regression model of infection revealed that men had lower odds of infection than women (OR = 0.72, $CI_{95}$ = [0.50, 1.01]) and participants from both Village A and Village S had twice the odds of infection compared to Village M (OR = 2.16, $CI_{95}$[1.42,3.29], OR=2.31, $CI_{95}$[1.43,3.75]; Table 2). In tests of our first and second hypotheses, we did not find a significant effect of housing material, sum of goods and land, and animal interactions. In tests of our third hypothesis, respondents that reported handwashing with only water had 1.39 odds of infection ($CI_{95}$ = [0.98, 1.99]) compared to washing hands with soap and water, while zoonotic literacy was not significantly associated with *Blastocystis* infection. Next, we investigated whether any of these variables predicted ST richness by fitting a Poisson GLM to the number of different *Blastocystis* subtypes that were isolated from each person (Table 2). We only identified support for an effect of village, with no statistical support for the variables predicted by our hypotheses. The effect was the same as in the main analysis, with higher ST richness in Villages A (RR = 1.41, $CI_{95}$[1.18,1.71]), and S (RR = 1.45, $CI_{95}$[1.18,1.79]), compared to Village M.

Finally, we investigated the predictors of infection with each subtype of *Blastocystis* (Table 3). Washing hands with only water, as compared to soap and water, increased the risk of infection for ST1 (OR = 1.81, $CI_{95}$[1.10,3.01]) and ST2 (OR = 1.70, $CI_{95}$[1.01,2.89]). In addition, males had lower risk (OR = 0.52, $CI_{95}$[0.33,0.80]) of infection than females for ST3. Age had a marginal effect in increasing the odds of infection for ST3. We also identified that Village A and Village S had higher odds of infection compared to Village M for all three subtypes. None of the other variables were statistically significant.

 

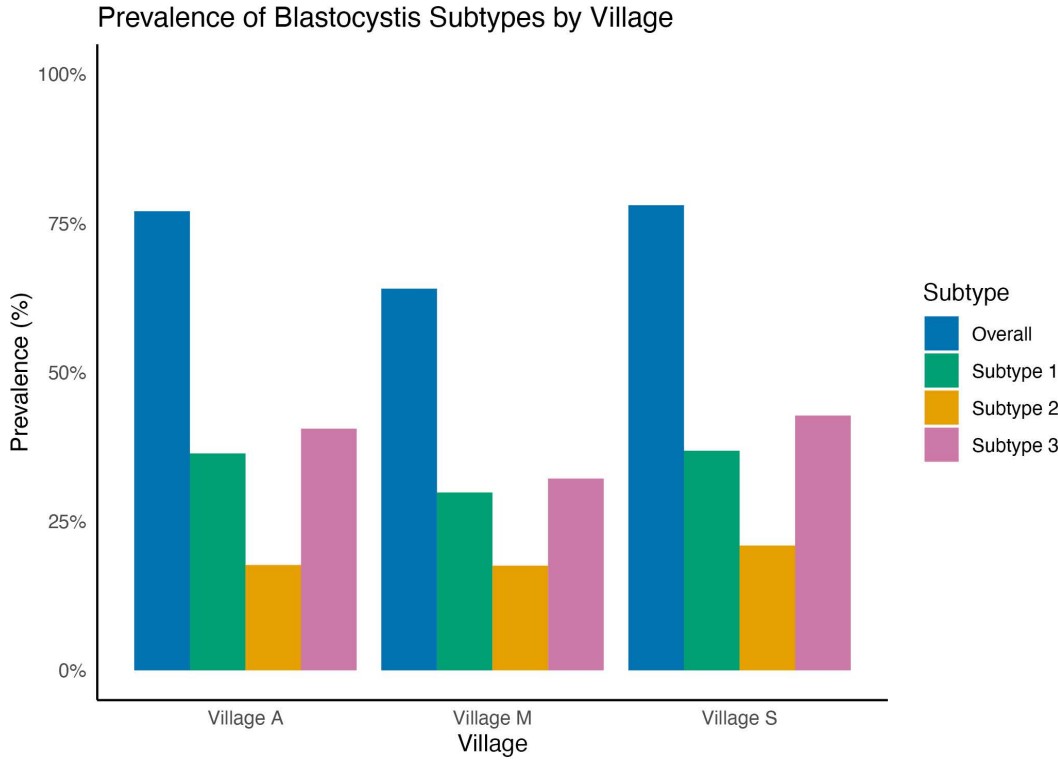

**Fig 1. Prevalence (%) of *Blastocystis* infection and prevalence of infection by subtype (ST) 1-3 by village.** Prevalence of participants who were positive for *Blastocystis* infection across all subtypes (ST) indicated by village. Additionally, the prevalence of participants positive for ST1, ST2, and ST3 infection is reported for each village. Infection status was determined using DNA metabarcoding of fecal samples, and subtype was identified using GenBank database.

**Table 2. GLM results for *Blastocystis* binomial infection (1,0) and richness (sum of ST's).**

| Variable | Binomial Infection Odds Ratio [95% CI] | ST Richness Rate Ratio [95% CI] |
|---|---|---|
| Intercept | 0.98[0.32,3.03] | 0.66[0.39,1.11] |
| Male | 0.72[0.50,1.01] † | 0.95[0.81,1.11] |
| Age | 1.01[1.00,1.02] | 1.00[1.00,1.01] |
| Education Level > *Primary Education* | 1.13[0.78,1.64] | 1.07[0.91,1.26] |
| Housing Material | 0.99[0.90,1.09] | 0.98[0.94,1.03] |
| Sum of Goods Owned | 1.05[0.93,1.19] | 1.02[0.97,1.08] |
| Handwashing without Soap | 1.39[0.98,1.99] † | 1.11[0.95,1.30] |
| Zoonotic Literacy | 0.90[0.60,1.34] | 0.95[0.80,1.14] |
| Animal Interactions | 1.03[0.97,1.10] | 1.02[0.99,1.04] |
| Animal Richness | 0.98[0.81,1.18] | 0.97[0.89,1.06] |
| Village A | 2.16[1.42,3.29] * | 1.23[1.01,1.52] * |
| Village S | 2.31[1.43,3.75] * | 1.34[1.07,1.68] * |

*p ≤ 0.05; † p<0.1.

**Table 3. GLM results for infection of ST1, ST2, and ST3.**

|  | ST1 (n = 138) | ST2 (n = 103) | ST3 (n = 173) |
|---|---|---|---|
| Variable | Odds Ratio [95% CI] | Odds Ratio [95% CI] | Odds Ratio [95% CI] |
| Intercept | 0.40[0.08,1.97] † | 0.34[0.06,1.96] | 0.21 [0.04,0.97] |
| Male | 0.78(0.49,1.26] | 1.09[0.65,1.82] | 0.52[0.33,0.80] * |
| Age | 1.00[0.98,1.01] | 0.99[0.97,1.00] | 1.01[1.00,1.03] † |
| Education Level >Primary Education | 0.93(0.56,1.53] | 0.88[0.50,1.54] | 0.98[0.61,1.57] |
| Housing Material | 1.03[0.90,1.18] | 0.98[0.84,1.14] | 1.03 [0.91,1.17] |
| Sum of Goods Owed | 1.03[0.87,1.21] | 1.13[0.95,1.35] | 1.07(0.92,1.35] |
| Handwashing without soap | 1.86[1.16, 2.99] * | 1.70[1.01,2.89] * | 1.09[0.69,1.72] |
| Zoonoses Literacy | 0.73[0.44,1.22] | 0.90[0.50,1.64] | 1.32[0.78,2.25] |
| Animal Interactions | 0.84[0.69,1.02] | 0.92[0.74,1.15] | 0.87[0.72,1.05] |
| Animal Richness | 1.18[0.97,1.45] | 1.11[0.90,1.38] | 1.12[0.92,1.35] |
| Village A | 2.18[1.24,3.94] * | 1.59[0.85,3.05] | 2.23[1.27,3.97] * |
| Village S | 1.99[1.03,3.89] * | 1.93[0.94,4.04] † | 2.81[1.51,5.35] * |

*p ≤ 0.05; † p < 0.1.

## Discussion

We applied the One Health Disparities (OHD) framework to investigate the predictors of *Blastocystis* infection among rural farmers in northeast Madagascar. This framework expands on the One Health approach – i.e., the interconnectedness of human, nonhuman animal, and environmental health – by incorporating the human social environment. This integrative perspective helped disentangle the underlying social determinants of how infection patterns varied by gender, handwashing practices, and village. However, OHD was less informative for understanding the role of wealth and animal contact. The findings highlight that, among the three villages studied, variation of behavior (gender norms and handwashing) and village, are key determinants of *Blastocystis* infection. Understanding these sociocultural influences is crucial for effectively reducing disease risk and providing village-level interventions.

Overall, we identified 76.5% overall infection prevalence of *Blastocystis* across the three villages, which is comparable to a study that identified 64.5% *Blastocystis* prevalence in an urban area of northwest Madagascar [40]. We tested *a priori* hypotheses related to wealth, animal interaction, and hygiene behaviors, along with relevant demographic variables, aiming to reveal the social factors that influence transmission, including differences among villages. Finally, we explored infection status by ST, focusing on individuals with only one ST to avoid any interactions that might occur among sub-types in co-infection. Through this stratified approach, we were able to identify handwashing and gender to be uniquely associated with different STs. We identified that both ST1 and ST2 were associated with increased odds of infection in people who reported washing their hands with only water. Females and older individuals had increased odds of ST3 infection. Notably, across all models we identified a strong effect for village.

First, we found no evidence supporting our wealth hypothesis, as neither wealth index variable significantly influenced the odds of infection. Our results align with a previous study that also did not identify commercial goods or household infrastructure as significant predictors of infection [26]. Future research could include additional wealth metrics, such as measures of agricultural wealth, as prior research has suggested that material goods asset assessments may be less effective at capturing wealth in rural settings [41–44]. It is also important to note that the population studied represents a relatively narrow range of wealth, with most individuals being relatively poor in a global context. This may indicate that the wealth gap relevant to influencing odds of infection may emerge at higher levels of wealth disparity. Incorporating

individuals from more diverse economic backgrounds in future studies could offer valuable insights into whether significant differences in infection odds occur within urban areas and between urban and rural populations.

We also failed to find an association between *Blastocystis* infection and contact with potential animal reservoirs [26]. To further understand the role of human-animal transmission of *Blastocystis*, future research could sample a subset of the most common animals for specific *Blastocystis* STs. For example, participants often reported interactions with poultry, cats, and dogs, as well as observing rodents inside their homes and consuming food in proximity to fecal matter (S1 Fig). A limitation of our analysis was that we analyzed the number of different types of interactions rather than the intensity of interactions, either in terms of frequency or number of specific animals of a given species with whom an individual interacts [45]. Since transmission is the product of contact rate and probability of infection given contact, a future metric could obtain data on the frequency of contact with animals and feces, along with details on the number of individual animals involved.

We found some associations between handwashing and infection status. Study participants who reported using only water, as compared to soap and water, to wash their hands had increased odds of infection for ST1 and ST2. Our results align with an ongoing issue throughout Madagascar. From 2000 to 2020, Madagascar has not seen any significant changes in access to clean water, with approximately 80% of the rural population having limited or no access to water for hygiene [46]. Hand washing with soap is essential for human health and is protective against GI and respiratory diseases [47]. However, low-income countries face inadequate access to soap and water [48,49]. Globally, approximately 2.3 billion people lack household handwashing facilities with soap and water [50]. Limited access to water and soap has been previously identified as a significant contributor to limited practice of effective hand hygiene in rural communities [50–52]. Future research could investigate barriers to accessing soap, in addition to increasing understanding about handwashing frequency, source of water used to practice hand hygiene, and pathogen occurrence in the water sources used to wash hands, dishes, and clothing.

Overall, we identified the highest infection rate of ST3 (39%), followed by ST1 (35%), consistent with other studies that report ST3 as being highly prevalent in humans, with ST1 being the second-most common [53]. This pattern was not consistent in Village A, where ST1 had a slightly higher prevalence than ST3. Due to the high global prevalence of ST3 in humans, it is considered to be the most human-specific ST and is thought to predominantly transmit via human-to-human contact [53]. ST3 has been described to have less intra-subtype genetic diversity compared to other subtypes [54–56]. This reduced diversity may be attributed to its host specificity in humans, while a greater diversity in ST1 and ST2 may be explained by their broader host range and varied transmission modes. This may explain the observed gender difference in ST3 infection, with females showing a higher risk of infection than males. This disparity could be attributed to unmeasured gender roles, such as caregiving responsibilities, which may increase human-to-human transmission of *Blastocystis* [57,58]. Finally, we identified that Villages A and S had significantly higher odds of infection in comparison to Village M, which may be due to variation in housing proximity, population density, or water source [26].

In conclusion, we identified a high prevalence of infection with the protozoan *Blastocystis* in rural Malagasy farmers. Our findings provide further evidence about the risk of microbial transmission pathways in rural farmers via inadequate hand hygiene practices, but less evidence for effects of interspecies interactions. Investigating the determinants of disease exposure is crucial because rural populations face severe health inequities exacerbated by limited access to clean water and basic sanitation services. Our findings suggest that access to clean water and hygiene products play a critical role in reducing the risk of *Blastocystis* infection in this region of Madagascar. In addition, a social network analysis in the same villages found that shared environments, within villages, predicted infection risk of *Blastocystis*, likely due to environmental transmission through shared water sources [26]. By leveraging the OHD framework, future studies could further explore the social and systemic barriers that restrict access to clean water and sanitation in rural areas. Notably, ensuring clean water and sanitation aligns with Sustainable Development Goal 6 of the United Nations. In 2022, Madagascar received $2.2 million in National Water Project funding from The World Bank to increase access to clean water in the capital city Antananarivo and surrounding areas [59]. However, additional efforts are needed to address the challenges highlighted by our study in rural regions globally.

## Supporting information

**S1 Fig. Reported animal contact across 11 animals and 8 types of interactions from all three villages.** The total reported interactions with 11 animal species (dogs, cats, pigs, cows, poultry, carnivores, goats/sheep, rodents, bush pigs, wild birds, and tenrecs) and the total number of 8 interaction types (e.g., consumed when sick, consumed raw/under-cooked, feces near food, shared water, raised, handled, kept inside dwelling, and kept as pets). "NA" indicates no reported interaction for a specific animal species.
(TIF)

**S2 Fig. Scored (1–4) household material used for floor, roof, and walls across all three villages.** Reported materials used for household outer walls, roofs, and floors were assessed through three questions with identical response options: thatch, compacted earth, burnt brick, metal sheets (iron, aluminum), mud, mud brick, concrete/cement, bamboo, wood planks, travelers palm, and raffia palm. Each material was ranked numerically from 1 to 4 based on local availability and cost. The scores from the three categories were summed to calculate a housing material index score, ranging from 3 to 12. Circle size and color represent the count/frequency of reported housing material scores.
(TIF)

**S1 Checklist. Inclusivity in global research.**
(PDF)

## Acknowledgments

We thank the Malagasy Ethics Panel for permission to conduct the research as well as the Duke Lemur Center-SAVA, and Association Vahatra team for their invaluable logistical support. We also greatly appreciate the participation and hospitality of members of the three study communities.

## Author contributions

**Conceptualization:** Alma Solis, Georgia Titcomb, Charles L. Nunn.

**Data curation:** Georgia Titcomb, Kayla Kauffman, Michelle Pender.

**Formal analysis:** Alma Solis, Angela Anaeme, Georgia Titcomb, Mark Janko, Charles L. Nunn.

**Funding acquisition:** Voahangy Soarimalala, Randall Kramer, Hillary Young, Charles L. Nunn.

**Investigation:** Alma Solis, Georgia Titcomb, Mark Janko, Tyler M. Barrett, Kayla Kauffman, Voahangy Soarimalala, Lev Kolinski, Randall Kramer, Hillary Young, Charles L. Nunn.

**Methodology:** Alma Solis, Georgia Titcomb, Mark Janko, Jean Yves Rabezara, Kayla Kauffman, Voahangy Soarimalala, Randall Kramer, Hillary Young, Charles L. Nunn.

**Project administration:** Michelle Pender, Voahangy Soarimalala.

**Supervision:** Voahangy Soarimalala, Randall Kramer, Charles L. Nunn.

**Validation:** Georgia Titcomb.

**Visualization:** Alma Solis, Lev Kolinski.

**Writing – original draft:** Alma Solis.

**Writing – review & editing:** Angela Anaeme, Georgia Titcomb, Jean Yves Rabezara, Kayla Kauffman, Lev Kolinski, Hillary Young.

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
