## [Decision Letter · Decision Letter 0]

30 May 2025

PGPH-D-25-00246

One Health Disparities and Blastocystis infection among smallholder farmers in northeastern Madagascar

Dear Dr. Alma Solis,

Thank you for submitting your manuscript to PLOS Global Public Health. After careful consideration, we feel that it has merit but does not fully meet PLOS Global Public Health’s publication criteria as it currently stands. Therefore, we invite you to submit a revised version of the manuscript that addresses the points raised during the review process.

We look forward to receiving your revised manuscript.

Kind regards,

Muhammad Asaduzzaman, MD MPH MPhil

Academic Editor

Journal Requirements:

Additional Editor Comments (if provided):

Reviewers' comments:

Reviewer's Responses to Questions

**Comments to the Author**

1. Does this manuscript meet PLOS Global Public Health’s publication criteria?

Reviewer #1: Yes

Reviewer #2: Yes

2. Has the statistical analysis been performed appropriately and rigorously?

Reviewer #1: Yes

Reviewer #2: Yes

3. Have the authors made all data underlying the findings in their manuscript fully available (please refer to the Data Availability Statement at the start of the manuscript PDF file)?

Reviewer #1: No

Reviewer #2: Yes

4. Is the manuscript presented in an intelligible fashion and written in standard English?

Reviewer #1: Yes

Reviewer #2: Yes

Reviewer #1: The authors have not made all data underlying the findings available but have undertaken a comprehensive analysis using high level statistics and this is available in the manuscript. The manuscript is technically sound and the conclusions is supported by the data.

THe language is clear and there is no ambiguity except few typos that can be addressed.

Reviewer #2: 1. Please, clarify how the required sample size of participants enrolled in the study was estimated

2. Also, it is unclear from the method whether the same individuals sampled for the Blastocystis screening participated in the survey, although the presentation of the results shows so. Please, clarify in your methodology.

3. Please, resolve a few punctuation errors/omissions e.g as in line 182

4. Line 378: Please, include the intext citation for the appropriate reference

5. Line 419 - 427: Please, adjust inconsistent text font.

**Do you want your identity to be public for this peer review?** For information about this choice, including consent withdrawal, please see our Privacy Policy

Reviewer #1: No

Reviewer #2: **Yes: ** Marvellous O. Adeoye

---

## [Decision Letter · Decision Letter 1]

26 Aug 2025

One Health Disparities and Blastocystis infection among smallholder farmers in northeastern Madagascar

PGPH-D-25-00246R1

Dear Alma Solis,

We are pleased to inform you that your manuscript 'One Health Disparities and Blastocystis infection among smallholder farmers in northeastern Madagascar' has been provisionally accepted for publication in PLOS Global Public Health.

Best regards,

Muhammad Asaduzzaman, MD MPH MPhil

Academic Editor

Reviewer Comments (if any, and for reference):

Reviewer's Responses to Questions

**Comments to the Author**

Reviewer #1: All comments have been addressed

Reviewer #2: (No Response)

publication criteria?

Reviewer #1: Yes

Reviewer #2: Yes

3. Has the statistical analysis been performed appropriately and rigorously?

Reviewer #1: Yes

Reviewer #2: I don't know

4. Have the authors made all data underlying the findings in their manuscript fully available (please refer to the Data Availability Statement at the start of the manuscript PDF file)?

Reviewer #1: Yes

Reviewer #2: Yes

5. Is the manuscript presented in an intelligible fashion and written in standard English?

Reviewer #1: Yes

Reviewer #2: Yes

Reviewer #1: All comments raised has been addressed by the authors and the language is clear and unambiguous

Reviewer #2: 1. A quick review of punctuation would improve readability (e.g., lines 86, 101, 136–137).

2. Statistical Analysis

Could you please clarify the rationale for applying GLM with snowball sampling? As snowball sampling is a non-probability method, it would be useful to explain how this impacts the reliability of the inferences drawn.

3. Consistent Vancouver Referencing Style

Please, revise references 46 and 59 to match Vancouver format (include full organisation names, publisher/location details, and page count/date as appropriate).

**Do you want your identity to be public for this peer review?** For information about this choice, including consent withdrawal, please see our Privacy Policy

Reviewer #1: No

Reviewer #2: **Yes: ** Marvellous O. Adeoye
